# Sex and age differences in the association between routine suicide newspaper reporting and change in admissions of suicidal patients: An investigation at an emergency and critical care center in Tokyo

Yasushi Emura[1,2⊜], Sho Kanata[1⊜], Naoki Hayashi[1,3*], Ken-ichi Matsumura[1‡], Akihisa Akahane[1‡], Mamoru Tochigi[1,4], Hiroshi Kunugi[1]

1 Department of Neuropsychiatry, Teikyo University School of Medicine, Graduate School of Medicine, Tokyo, Japan, 2 Emura Mental and Internal Medicine Hospital, Nemuro, Japan, 3 Nishigahara Hospital, Tokyo, Japan, 4 Health Care Center, the University of Electro-Communications, Tokyo, Japan

⊜ These authors contributed equally to this work.
‡ KM and AA also contributed equally to this work.
* nhayashi55@nifty.com

## Abstract:

### Background

The influence of media reporting on suicide and suicide attempts (SA) has been rigorously studied. Previous studies focused on characteristics like celebrity status, sex and age of suicide decedents, which may facilitate suicide and SA occurrence. These studies have informed guidelines for responsible media reporting. However, the nuanced effects on different sex and age groups of suicidal individuals remain less understood.

### Methods

This study examined the association between the characteristics of initial suicide reports in four major newspapers and the differences in admission numbers (Δs) of SA patients across sex and age groups during pre- and post-article release one-week periods. Data from an Emergency and Critical Care Center from 2012 to 2019 were obtained through a review of medical records. Nonparametric MANOVAs were employed to investigate Δs for sex and age groups of SA patients in relation to sex and age of suicide decedents, incident types, and methods. Significant differences were evaluated using Wilcoxon tests.

### Results

A total of 1,205 articles on 676 suicide incidents and 1,081 SA admissions were analyzed. MANOVAs revealed a significant association between the Δs and the reported

**Data availability statement:** All relevant data are within the manuscript and its Supporting Information files.

**Funding:** The author(s) received no specific funding for this work.

**Competing interests:** The authors have declared that no competing interests exist.

suicide methods. Δ for females was positively associated with reports of gas poisoning and other infrequently reported methods, whereas Δ for males was negatively associated with gas poisoning. Δ for younger patients was positively associated with infrequently reported methods and negatively with firearm discharge. No significant associations were found between Δ and sex and age of decedents or incident types.

## Discussion

This study demonstrates a differential impact of suicide news articles reporting gas poisoning, infrequently reported suicide methods, and firearm discharge, on SA patient admissions across sex and age groups. These differences may arise from variations in emotional responses to the suicide methods, which can be understood in both psychological and socio-cultural contexts. Further research is needed to clarify the determinants of differential influences to more effectively address and mitigate the risks of suicide and SAs.

## Introduction

Suicide attempts (SAs) represent a significant global public health concern by imposing a substantial burden on emergency services, creating complex therapeutic challenges for mental health professionals, and highlighting the essential societal responsibility of suicide prevention. To address these challenges, extensive efforts have been made to identify the underlying factors, including psychiatric issues, socio-cultural influences, and economic conditions [1]. Among these factors, the harmful effects of media activities that may induce imitation of reported behavior, referred to as the 'Werther effect,' were established in previous studies [2–7]. Research has further highlighted the central role of factors such as the quantity of coverage and the emotional impact of media activities in the harmful effects [2,4–6,8–15]. On the other hand, protective effects of media articles, such as the introduction of positive coping strategies, have also been reported, which are known as the 'Papageno effect' [8].

Driven by these research achievements, a number of mental health institutes, including the World Health Organization (WHO), have issued guidelines for media professionals to responsibly deliver accurate and appropriate messages while being cautious to avoid harmful effects [16,17]. Emerging studies have begun to indicate a reduction in the harmful impact on suicide following the promulgation of the guidelines [16,18,19].

Apart from the role of media messages in imitative suicide and SA, the characteristics of recipient populations who are susceptible to these influences have also been an important focus of investigation [13,15]. Previous studies have specified various populations susceptible to imitative suicide or SA, such as younger persons [20,21], middle-aged persons [22], elderly persons, [23] females [2] and males. [24] Though the results remain inconclusive, Domaradzki's review [15] suggested that younger persons and females are more susceptible to media influence.

Imitative suicide and SA following media reporting are believed to involve interactive processes between the reporting of suicide and the individuals who subsequently attempt suicide. [6,11] Beyond more general concepts of suggestion [2] and modelling, [5] identification and social learning theories [4,6,13,15] have been used to explain this imitation. Identification may lead recipients to engage in similar behavior, whereas social learning theory suggests that individuals in distress may perceive suicide as a solution through media messaging. These theories have been reinforced by studies demonstrating that suicidal individuals often exhibit characteristics similar to those of suicide decedents, particularly in terms of age, sex, suicide method, and life situations, as discussed in reviews [6,11,13,15,25] and meta analyses [26,27].

However, it is notable that most of these findings have been derived from research on suicide rather than SA, making their direct application to SA imitation potentially problematic. Differences in responses to media suicide reports between suicide and SAs have been observed. Suicide attempters tend to respond more sensitively to reported decedents who share similar characteristics with them [28] and more quickly to media exposure than those who die by suicide [29]. Furthermore, most studies addressing factors associated with SA have focused on a small number of anecdotal suicide incidents and often involved a limited range of suicide types. Consequently, further clarification of the relationship between media suicide reporting and recipients of those message who subsequently attempt suicide remains an important focus of investigation.

The aim of this study was to explore the relationship between newspaper initial reports of suicide incidents and hospital admissions of SA patients immediately following the release of the report. Specifically, this study aimed to examine the influence of basic characteristics of newspaper suicide reports on the admissions of SA patients across sex and age groups. A notable distinction of this study is that it primarily investigated differences in sex and age groups in relation to routine suicide reports, whereas previous research has largely focused on a limited number of specific incidents, such as celebrity suicides or other high-profile cases. The findings of this study are expected to enhance our understanding of media influence on suicidal individuals and offer insights that can be applied in the prevention and treatment for suicidal individuals particularly susceptible to media reports.

## Methods

### Newspaper data

The present study utilized the news article databases provided by the four largest nationwide daily newspapers in Japan: Yomiuri, Asahi, Mainichi, and Sankei (listed in the order of their circulation numbers). These newspapers accounted for over 50% of the national market share, with an even higher share in Tokyo [30]. Japan has a high newspaper circulation compared with other countries, with newspapers holding a significant share of the overall media landscape (38–40 million in circulation during the study period). In 2015, approximately the midpoint of the study period, the newspaper subscription rate in Japan was reported to be 77.7% across all age groups and 40.5% among those in their 30s [31]. Their news articles are also frequently used as sources for internet news.

Data on suicide incident reports were retrieved from both national and local editions, and morning and evening editions, distributed in the northwest region of Tokyo's main districts between April 7, 2012, and January 25, 2019. A digital search was conducted across the full text of articles using the search terms "Jisatsu (suicide)" and "Shinju (homicide-suicide or multiple suicide that occurred in particularly intimate relationships)." "Shinju" is a term unique to Japan and has long been a theme in Japanese theater, literature, and other cultural expressions [32].

To examine the association between the characteristics of initial suicide reports immediately after incidents and the short-term change in SA patient admissions, we selected articles reporting actual suicides within a week of publication, excluding follow-up reports.

To investigate article characteristics, we chose items of the sex and age of suicide decedents, which are central to the focus of this study, their social status, occupation, and the suicide methods used. Incident characteristics examined

included homicide-suicide (cases involving the murder of another person prior to the suicide) and multiple-suicide (cases involving more than one decedent, including serial or group suicides) as prominent features of suicide. On the other hand, characteristics that tend to vary in article content across media types and time periods, such as the background, reasons, or consequences of the incident, were not included.

In cases where the age was described only approximately in the articles, a representative value was assigned. For example, when a decedent was referred to as being in their "forties," it was recorded as "45 years old." Social status or occupation was classified into the following categories in the assessment: Famous person (including television talent, professional athlete, company manager, or politician), Public worker, Professional expert, Police officer or Self-Defense Force (SDF) member, Criminal or suspected person, and Other occupation or not reported. Assessed suicide methods included Hanging, Railroad jumping, Jumping from a height (Height jumping), Gas poisoning, Firearm discharge, Burning, and Cutting or stabbing. For analysis, the items used were the most frequently observed methods among reported suicide decedents, Other methods, and No method reported.

To present the characteristics of reported suicide incidents and suicide decedents separately, two distinct datasets were created, with duplication from multiple newspaper reports on the same incident removed. Specifically, data from the earliest published article were selected; furthermore, if multiple reports on the same incident were published on the same day, the report from the newspaper with the largest circulation was used.

Incident types primarily consisted of Famous person suicide, Homicide-suicide, and Multiple suicide. Incidents that did not fall under any of these types were categorized as unclassified type. When incident types were treated as a single categorical variable, those that fell under more than one type were classified as combination type.

Regarding suicide news articles, minor adjustments were made to prepare the data for analysis. To accurately rate incidents involving more than one suicide decedent, 2-value (0, 1) variables related to sex and age—Female decedent included and Younger decedent included (defined as a decedent equal to or younger than the median age)—were created. When suicide decedents used different suicide methods within an incident, the less frequent one was used operationally for classification. Additionally, to make a concise assessment of whether or not an article contained descriptions to be avoided (as defined in the WHO recommendation Update, 2008), six 2-value items were included in the investigation: 1. Implying a simple reason for suicide; 2. Glorifying or sensationalizing; 3. Mentioning religious or cultural stereotypes; 4. Blaming or criticizing; 5. Adding special meaning, such as describing it as the first one ever seen or taking place in a well-known suicide spot; and 6. Referring to the involvement of mental disorders.

### Data of SA patients

SA patients were individuals hospitalized for the treatment of physical injuries resulting from suicide attempts, which are defined as concrete actions based on suicidal ideation [33], and who received psychiatric liaison services at the Emergency and Critical Care (ECC) Center of Teikyo University Hospital. This hospital is a leading emergency medical facility in the north-west region of the Tokyo metropolitan area. During the study period, the hospital handled over 12,000 emergency cases annually, including emergency room walk-in patients, and hospitalized approximately 2,300 patients each year [34].

Data on SA patients admitted to the ECC Center from April 1, 2012, to January 31, 2019, were accessed through the liaison service data and retrospectively reviewed using medical records. These included demographics such as sex, age, occupational and marital status, SA method, and psychiatric diagnosis based on the International Classification of Diseases, 10th revision (ICD-10). Physical damage (medical risk) caused by SA and the intention of suicide (suicidality) at the time of SA were assessed using items 246 and 250 of the Schedule for Affective Disorders and Schizophrenia (SADS), respectively [35].

Admitted patients were grouped based on their sex and age: Female and Male groups, and Younger (equal to or less than the median age of reported suicide decedents) and Older (greater than the median age) groups.

## Statistical analyses

The analysis primarily focused on the difference (Δ) in admission numbers of SA patients during the one-week period starting from the day of incident news article release (defined as day 0), compared with the pre-release one week (day −7 to day −1). The articles for investigation were selected based on the condition that during the pre- and post-release 14 days (day −7 to day 6), either admission data was accessible or the count was zero. This approach principally followed the methodology applied by Sinyor et al [23]. However, unlike the original study, which set a one-week lag period between the pre-release control period and the post-release period to account for multiple reports on the same incident across various media types, this study analyzed only the initial report for each newspaper, rendering the lag period unnecessary. Δs for the whole patient group and for each of the sex and age groups were labeled as: ΔWhole, ΔFemale, ΔMale, ΔYounger, and ΔOlder.

The preliminary stage of the analysis involved examining differences in sex and age group associations in the characteristics of reported suicide decedents and incidents, and admitted SA patients using the χ² tests.

This included assessing changes in the number of admission numbers of sex and age patient groups between the pre- and post-article release one-week periods with the Wilcoxon signed-rank tests, and performing correlation analyses by calculating Spearman rank-order correlation coefficients among Δs for patient groups, three component article characteristics: sex- and age-related characteristics (e.g., inclusion of female decedents and younger decedents), article types (i.e., Famous person suicides, Homicide-suicides, and Multiple suicides), and Suicide methods.

The main analysis involved examining the associations of Δs for sex and age patient groups (dependent variables) with each of the three components of article characteristics (independent variables) using nonparametric MANOVAs developed by Friedrich [34]. Additionally, comparisons of Δs for the patient groups, classified by article characteristics, were conducted with Δ for the reference group using Wilcoxon rank order tests. The nonparametric MANOVA model was crucial for evaluating the associations while accounting for the interrelationships among the dependent variables. More importantly, it resolved the issue of matrix singularity among these variables in the analysis.

The statistical software R (version 4.4.3, R Core Team 2025) was used to perform all statistical analyses. Nonparametric MANOVAs were conducted using the MANOVA.WIDE function (10,000 bootstrap samples) included in the package MANOVA.RM (version 0.5.4). Effect size r for Wilcoxon rank-sum test was calculated using wilcoxonR function in the Package rcompanion (version 2.5.0).

For the analyses of news articles, since each incident was covered in slightly fewer than two articles on average—effectively resulting in repeated testing—the significance level was set at 0.025 (0.05/2) (two-tailed) using Bonferroni's correction.

For the analyses of data with duplication-eliminated incident cases, a significance level of 0.05 was applied.

## Ethics approval

This research was conducted with the approval of the Teikyo University Ethical Review Board (Approval No. 19−003). The study was based on publicly available newspaper data and a retrospective review of medical records of the subject patients. For the part of medical record review, where patient consent was waived, an opt-out method was employed by posting the study overview within the hospital from April 5, 2019, until December 31, 2021. Patients who expressed their intention not to participate were excluded from the study as an ethical consideration. Access to medical records for research purposes was also restricted only during the posting period.

## Results

### Characteristics of newspaper articles

A total of 1,205 news articles covering 676 suicide incidents were published by the four major newspapers in the region during the study period. The numbers of articles published by Yomiuri, Asahi, Mainichi, and Sankei were 334, 186, 286,

and 399, respectively. The toll of suicide decedents in reported incidents was 771 deaths. Among the incidents, 299 (44.2%) were covered by more than one newspaper: 135 (20.0%) by two newspapers, 98 (14.5%) by three newspapers, and 66 (9.8%) by four newspapers. The toll of suicide decedents across the 676 incidents was as follows: 606 incidents (89.6%) involved a single decedent, 53 incidents (8.7%) involved two decedents, 12 incidents (1.8%) involved three decedents, three incidents (0.4%) involved four decedents, and one incident each (0.1%) involved five and six decedents.

Table 1 presents the characteristics of the reported suicide decedents, including age distribution, social status or occupation, and suicide methods by sex and age groups. The sexes of the suicide decedents were 531 males, 226 females, and 14 unclear. The mean age (SD, median) of 666 decedents with reported age was 37.2 (20.9, 35) years; males 38.5 (20.7, 36) and females 34.4 (20.9, 30.5).

Significant differences by sex and age group were found, as follows: younger decedents (≤ 35 years) included a higher percentage of females; females were less represented among occupational or social status groups of Famous persons, Public workers, Police officers or SDF members, and Criminal or suspected persons, as well as in cases involving Hanging or Firearm discharge; in contrast, females were more represented in cases involving Height jumping and Gas poisoning.

Younger decedents accounted for a smaller percentage of Famous persons, Public workers, and Professional experts; in contrast, they accounted for a higher percentage in cases involving Railroad jumping and Height jumping, and a lower percentage in cases involving Hanging, Firearm discharge, and Cutting or stabbing.

Table 2 presents the types of suicide incidents along with their sex- and age-related characteristics. Significant associations between suicide incident types and sex- and age-related characteristics were observed, as follows: Female decedent included was negatively associated with Famous person suicide and positively associated with Multiple suicide; Younger decedent included was negatively associated with both Famous person suicide and Homicide-suicide.

Responses for items examining adherence to media recommendations were as follows: 1. Implying a simple reason for suicide, 91 (7.6%) (none provided definitive conclusions on the reasons); 2. Glorifying or sensationalizing, 0 (0.0%); 3. Mentioning religious or cultural stereotypes, 3 (0.2%); 4. Blaming or criticizing, 0 (0.0%); 5. Adding special meaning to the incident, 0 (0.0%); and 6. Referring to mental disorders, 3 (0.2%). Overall, we found that the article descriptions were restrained, and they largely adhered to WHO recommendations.

## Characteristics of SA patients

The SA sample consisted of 1,081 suicidal patients (692 females, 389 males) with a mean age (SD, median) of 38.8 (15.8, 36) years: females 37.9 (15.9, 35) and males 40.4 (15.5, 38). The female patients were significantly younger than male patients (p = 0.002, Wilcoxon test). Marital status was distributed as follows: unmarried, 500 (46.3%); ever married, 418 (38.7%); and unknown, 163 (15.1%). Employment status was: employed, 495 (45.8%); unemployed, 362 (33.5%); and unclear, 224 (20.7%). The most frequent ICD-10 diagnoses were Mood disorders (F3), 583 (53.9%); Schizophrenia, schizotypal, and delusional disorders (F2), 178 (16.5%); Neurotic, stress-related, and somatoform disorders (F4), 128 (11.8%); Disorders of adult personality and behavior (F6), 82 (7.6%); and Mental and behavioral disorders due to psychiatric substance use (F1), 65 (6.0%). The most frequent SA methods were: Overdosing on prescribed or OTC medicine, 728 (67.3%); Height jumping, 164 (15.2%); Cutting or stabbing, 156 (14.4%); and Hanging, 52 (4.8%).

Table 3 shows the sociodemographic and clinical characteristics of the patients by sex and age groups. The percentages of patients with higher than moderate medical risk due to SA (SADS Item 246) and seriousness of suicidal intent (SADS Item 250) were significantly higher among male patients and older (> 35 years) ones than female ones and younger (≤ 35 years) ones, respectively.

Significant sex and age group differences in SA methods were observed, as follows: the percentage of Overdosing was higher among female patients than among male patients, and higher among younger patients than among older patients;

**Table 1. Characteristics of the reported suicide decedents by sex and age.**

| | Male decedents (N=531) (N (%)) [a] | Female decedents (N=226) (N (%)) [a] | Younger decedents (N=349) (N (%)) [b] | Older decedents (N=318) (N (%)) [b] | Total (N=771) (N (%)) |
|---|---|---|---|---|---|
| Age distribution of suicide decedents [c] | | | | | |
| 11-21 years | 132 (24.9) | 83 (36.7) | | | 215(27.9) |
| 21-35 years | 98 (18.5) | 36 (15.9) | | | 134(17.4) |
| 36-50 years | 85 (16.0) | 41 (18.1) | | | 126(16.3) |
| 51-65 years | 95 (17.9) | 27 (11.9) | | | 122(15.8) |
| 66-80 years | 42 (7.9) | 10 (4.4) | | | 57(7.4) |
| 81-98 years | 10 (1.9) | 8 (3.5) | | | 18(2.3) |
| Not reported | 69 (13.0) | 21 (9.3) | | | 104(13.5) |
| Social status or occupation | | | | | |
| Famous person | 48 (9.0)*** | 4 (1.8) | 8 (2.3) | 38 (11.9)*** | 52 (6.7%) |
| Public worker | 33 (6.2)** | 3 (1.3) | 9 (2.6) | 19 (6.0)* | 40 (5.2%) |
| Professional expert | 9 (1.7) | 1 (0.4) | 0 (0.0) | 10 (3.1) | 10 (1.3%) |
| Police officer or SDF member | 38 (7.2)*** | 0 (0.0) | 13 (3.7) | 21 (6.6) | 39 (5.1%) |
| Criminal or suspected person | 27 (5.1)* | 2 (0.9) | 10 (2.9) | 18 (5.7) | 29 (3.8%) |
| Other occupation or not reported | 376 (70.8) | 216 (95.6)*** | 309 (88.5)*** | 212 (66.7) | 601 (78.0%) |
| Suicide method | | | | | |
| Hanging | 119 (22.4)** | 29 (12.8) | 53 (15.2) | 90 (28.3)*** | 148 (19.2%) |
| Railroad jumping | 97 (18.3) | 51 (22.6) | 89 (25.5)*** | 29 (9.1) | 149 (19.3%) |
| Height jumping | 86 (16.2) | 58 (25.7)** | 102 (29.2)*** | 38 (11.9) | 146 (18.9%) |
| Gas poisoning [d] | 35 (6.6) | 34 (15.0)*** | 31 (8.9) | 29 (9.1) | 72 (9.3%) |
| Firearm discharge [e] | 66 (12.4)*** | 1 (0.4) | 19 (5.4) | 36 (11.3)** | 68 (8.8%) |
| Burning | 32 (6.0) | 16 (7.1) | 20 (5.7) | 13 (4.1) | 55 (7.1%) |
| Cutting or stabbing | 18 (3.4) | 8 (3.5) | 4 (1.1) | 20 (6.3)*** | 26 (3.4%) |
| Other methods [f] | 19 (3.6) | 14 (6.2) | 10 (2.9) | 19 (6.0)** | 33 (4.3%) |
| No method reported | 59 (11.1) | 15 (6.6) | 21 (6.0) | 44 (13.8)** | 74 (9.6%) |

Note

When multiple newspapers reported on the same incident, to avoid duplication, the suicide decedents presented in this table were those described in either the earliest released article or, in cases of simultaneous reporting, the one from the newspaper with the largest circulation.

[a]Fourteen decedents whose sex were not reported were not included in the calculation.

[b]Younger and Older patient groups were defined as those with age of equal to or less than 35 years (≤ 35 years) and those with age greater than 35 years (> 35 years), respectively.

[c]One hundred and four decedents whose age were not reported were not included in the count.

The association between Sex and age group were not significant ($\chi^2$ (1) = 3.564, 232/462 vs 86/205, p=0.059)

[d]Of the suicides by gas poisoning, 64 cases (88.9%) involved charcoal burning (CO poisoning).

[e]Among the suicides by firearm discharge, 15 cases (22.1%) were committed by police officers or Self-Defense Forces (SDF) members.

[f]Other methods included infrequently reported suicide methods, such as drowning, 19; suffocation, 4; explosion, 3; and overdosing or poison-taking, 3.

The asterisks indicate that the corresponding values are significantly higher than those of the respective sex and age groups. Significance levels: * p<0.05, ** p<0.01, *** p<0.001.

the percentages of Height jumping and Hanging were higher among male patients than among female patients; the percentage of Cutting or stabbing was higher among male patients than among female patients, and higher among older patients than among younger patients.

**Table 2. Suicide incident types by sex and age-related characteristics.**

| Incident types | Female decedents included (N=204) (N (%)) | Younger decedents included (N=322) (N (%)) | Total (N=676) (N (%)) |
|---|---|---|---|
| Famous person suicide | 4 (2.0)*** | 8 (2.5)*** | 51 (7.5) |
| Homicide-suicide | 61 (29.9) | 52 (16.1)*** | 195 (28.8) |
| Multiple suicide | 60 (29.4)*** | 36 (11.2) | 70 (10.4) |
| Non-classified | 118 (57.8) | 183 (56.8) | 377(55.8) |

Note

When multiple newspapers reported on the same incident, to avoid duplication, the suicide incidents presented in this table were those described in either the earliest released article or, in cases of simultaneous reporting, the one from the newspaper with the largest circulation.

Non-classified: suicide not classified into Famous person suicide, Homicide suicide or Multiple suicide.

Significance levels of the association: * p<0.05, ** p<0.01, *** p<0.001.

**Table 3. Demographic and clinical characteristics of the suicidal patients by sex and age.**

| | Female (N=692) N (%) | Male (N=389) N (%) | Younger patients [a] (N=532) N (%) | Older Patients [a] (N=546) N (%) | Total Patients (N=1081) N (%) |
|---|---|---|---|---|---|
| Age distribution at admission [b] | | | | | |
| 10-20 years | 69 (10.0) | 29 (7.5) | | | 98 (9.1) |
| 21-35 years | 298 (43.1) | 136 (35.1) | | | 434 (40.3) |
| 36-50 years | 192 (27.8) | 125 (32.3) | | | 317 (29.4) |
| 51-65 years | 76 (11.0) | 71 (18.3) | | | 147 (13.6) |
| 66-80 years | 46 (6.7) | 21 (5.4) | | | 67 (6.2) |
| 81-91 years | 10 (1.5) | 5 (1.3) | | | 15 (1.4) |
| Medical risk by suicidal behavior: Higher than moderate (SADS Item 246) | 139 (20.1) | 132 (33.9)*** | 109 (20.5) | 162 (29.7)*** | 271 (25.1) |
| Seriousness of suicidal intent (suicidality): Higher than moderate (SADS Item 250) | 398 (57.5) | 256 (65.8)** | 293 (55.1) | 361 (66.1)*** | 654 (60.5) |
| Method of suicide attempt | | | | | |
| Overdosing [b] | 515 (74.4)*** | 213 (54.8) | 393 (73.9)*** | 333 (61.0) | 728 (67.3) |
| Height jumping | 91 (13.2) | 73 (18.8)* | 93 (17.5) | 71 (13.0) | 164 (15.2 |
| Cutting or stabbing | 84 (12.1) | 72 (18.5)** | 39 (7.3) | 117 (21.4)*** | 156 (14.4) |
| Hanging | 26 (3.8) | 26 (6.7)* | 21 (3.9) | 31 (5.7) | 52 (4.8) |

Note:

[a]Younger and Older patient groups were those with age of equal to or less than 35 years (≤ 35 years) and those with age greater than 35 years (> 35 years), respectively.

[b]One female patient and two male patients whose ages were not reported were not included in the count.

Overdosing: Overdosing of prescribed medicine or OTC drugs.

SADS: Schedule for Affective Disorders and Schizophrenia [35].

The association between sex and age groups was significant ($\chi^2$(1) = 10.476, 367/691 vs. 165/387, p=0.001).

The asterisks indicate that the corresponding values are significantly higher than those of the respective sex and age groups: * p<0.05, ** p<0.01, *** p<0.001.

## Changes in admission numbers for patient groups between Pre- and post-article release periods

The means (SD, median) of admission numbers during the pre- and post-article release one-week periods for the whole patient group as well as for the Female, Male, Younger, and Older patient groups were as follows: 3.007 (1.905, 3) and 3.365 (1.982, 3); 1.952 (1.502, 2) and 2.165 (1.594, 2); 1.056 (1.101, 1) and 1.200 (1.100, 1); 1.727 (1.465, 1) and 1.900 (1.463, 2); and 1.280 (1.148, 1) and 1.466 (1.284, 1), respectively. The significance of the increase in admission numbers across all patient groups far exceeded the designated level ($p < 0.025$) ($p < 0.001$ for the first four groups, and $p = 0.002$ for Older patient group).

## Correlation analyses of differences (Δs) in admission numbers for patient groups and news characteristics

Table 4 shows a correlation matrix of Δs for the whole patients, sex and age patient groups, and news characteristics. Weak but significant correlations ($p < 0.025$) between Δs for patient groups and article characteristics were as follows: ΔWhole had a negative correlation with Younger decedent included and Firearm discharge, and a positive correlation with Other (infrequently reported) methods; ΔFemale had a negative correlation with Firearm discharge and a positive correlation with Other methods; ΔMale had a negative correlation with Gas poisoning and a positive correlation with Other methods; and ΔYounger had a negative correlation with Firearm discharge and a positive correlation with Other methods.

Several highly significant mild or moderate correlations were observed among article characteristics. Correlations with values greater than 0.200 were as follows: Female decedent included had a positive correlation with Multiple suicide and Gas poisoning; Younger decedent included had a negative correlation with Famous person suicide and Homicide-suicide, and a positive correlation with Height jumping; Multiple suicide had a positive correlation with Gas poisoning; and Homicide-suicide had a negative correlation with Railroad jumping and a positive correlation with Firearm discharge.

## Results of nonparametric MANOVAs and comparisons of Δs for patient groups classified by article characteristics

Nonparametric MANOVA of Δs for sex and age patient groups by sex and age-related article characteristics of Female decedent included and Younger decedent included, yielded a non-significant result that the modified ANOVA-Type Statistics (MATS) (df = 4) for Female decedent included and Younger decedent included were 3.371 (parametric bootstrap sampling [paramBS] $p = 0.460$) and 6.467 (paramBS $p = 0.182$), respectively.

Table 5 shows means (SD) of Δs for patient groups classified by sex and age-related article characteristics. Differences in Δs for the patient groups were compared with those for the group by Female decedent included 0 and Younger decedent included 0 (reference). There was only a trend-level difference ($p < 0.05$) in ΔFemale for the group by Female decedent included 0 and Younger decedent included 1, compared with that for the reference group.

Nonparametric MANOVA of Δs for sex and age patient groups by incident types, yielded a non-significant result that the modified ANOVA-Type Statistics (MATS) (df = 16) for the variable of incident type was 16.297 (paramBS $p = 0.450$).

Table 6 shows means (SD) of Δs for patient groups classified by incident types. Twenty-eight combination type incidents were Famous person suicide and Homicide-suicide, 7; Multiple suicide and Homicide-suicide, 17; and all 3 types, 4. There was no significant difference in Δ for any patient group by incident types, compared with Δ for the group by Unclassified type (reference). However, there was only a trend-level difference ($p < 0.05$) in ΔFemale for the group by Famous person suicide compared with that for the reference group.

Nonparametric MANOVA of Δ for sex and age patient groups with Suicide method yielded a highly significant result that the modified ANOVA-Type Statistics (MATS) (df = 32) for the variables of suicide method was 92.035 (paramBS $p < 0.001$). In the preparation of categorical variables for suicide methods, data from four articles that reported two suicide methods in a single incident were recoded to reflect the less frequent method.

Table 4. Correlation analyses of differences (Δs) in admission numbers for patient groups and news article characteristics.

| | ΔFemale | ΔMale | ΔYounger | ΔOlder | Female decedent (N=330) | Younger decedent (N=552) | Famous person (N=119) | Homicide-suicide (N=372) | Multiple suicide (N=122) | Hanging (N=256) | Height jumping (N=225) | Railroad jumping (N=209) | Firearm discharge (N=123) | Burning (N=74) | Gas poisoning (N=65) | Cutting or stabbing (N=42) | Other methods (N=47) |
|---|---|---|---|---|---|---|---|---|---|---|---|---|---|---|---|---|---|
| ΔWhole | 0.793*** | 0.539*** | 0.705*** | 0.645*** | -0.033 | -0.070* | 0.041 | -0.007 | 0.009 | -0.007 | 0.029 | -0.044 | -0.072* | 0.036 | 0.022 | 0.034 | 0.095*** |
| ΔFemale | | -0.044 | 0.582*** | 0.505*** | -0.039 | -0.060 | 0.042 | 0.007 | 0.041 | -0.026 | 0.048 | -0.054 | -0.067* | 0.051 | 0.059 | 0.014 | 0.097*** |
| ΔMale | | | 0.350*** | 0.376*** | -0.008 | -0.030 | -0.009 | -0.005 | -0.035 | 0.025 | -0.015 | 0.007 | -0.018 | -0.022 | -0.093** | 0.026 | 0.008 |
| ΔYounger | | | | -0.037 | -0.042 | -0.019 | 0.017 | 0.009 | 0.014 | -0.017 | 0.020 | -0.061 | -0.087** | 0.028 | 0.050 | 0.025 | 0.115*** |
| ΔOlder | | | | | 0.008 | -0.052 | -0.047 | -0.002 | 0.001 | -0.024 | 0.049 | 0.019 | -0.018 | 0.003 | -0.049 | -0.001 | 0.023 |
| Female decedent | | | | | | 0.096** | -0.128*** | 0.025 | 0.417*** | -0.087** | 0.073* | 0.068** | -0.189*** | -0.010 | 0.208*** | 0.056 | 0.059 |
| Younger decedent | | | | | | | -0.215*** | -0.211*** | 0.028 | -0.107** | 0.226*** | 0.181*** | -0.117*** | -0.069* | 0.068* | -0.093** | -0.082** |
| Famous person | | | | | | | | -0.155*** | -0.074* | 0.066* | -0.009 | -0.144*** | -0.075* | -0.073* | -0.042 | 0.043 | 0.048 |
| Multiple suicide | | | | | | | | | -0.099** | -0.067* | 0.009 | 0.052 | -0.104*** | 0.132*** | 0.334*** | 0.011 | 0.074* |
| Homicide-suicide | | | | | | | | | | -0.026 | -0.0623 | -0.244*** | 0.202*** | -0.044 | 0.023 | 0.186*** | 0.070* |

Note:

Δ: Difference in admission number for a patient group between pre- and post-article release one-week periods

ΔWhole: Δ for the whole patients, ΔFemale: Δ for female patients, ΔMale: Δ for male patients, ΔYounger: Δ for younger (≤ 35 years) patients, ΔOlder: Δ for older (> 35 years) patients

Female decedent: Article on an incident that included a female suicide decedent

Younger decedent: Article on an incident that included a younger (≤ 35 years) suicide decedent

Famous person: Article on an incident of famous person suicide

Height jumping: Jumping from a height

Significance level: * p < 0.025, ** p < 0.005, *** p < 0.0005

The correlation coefficients among Δs for patient groups were mostly moderate, with exceptions of those between ΔFemale and ΔMale, and between ΔYounger and ΔOlder.

**Table 5. Differences (Δs) in admission numbers for patient groups classified by reported suicide decedents' sex and age groups.**

| Articles: Female decedent included and Younger decedent included (N) | ΔFemale Mean (SD) | p-value[a] | ΔMale Mean (SD) | p-value[a] | ΔYounger Mean (SD) | p-value[a] | ΔOlder Mean (SD) | p-value[a] |
|---|---|---|---|---|---|---|---|---|
| 1, 1 (177) | 0.062 (2.198) | 0.074 | 0.062 (1.458) | 0.299 | −0.051 (1.853) | 0.147 | 0.175 (1.616) | 0.425 |
| 1, 0 (153) | 0.163 (1.819) | 0.103 | 0.255 (1.604) | 0.715 | 0.137 (1.899) | 0.415 | 0.281 (1.790) | 0.979 |
| 0, 1 (375) | 0.112(2.068) | 0.030 | 0.115 (1.616) | 0.662 | 0.189 (1.904) | 0.724 | 0.037 (1.847) | 0.070 |
| 0, 0 (500) | 0.358 (2.137) | | 0.162 (1.501) | | 0.250 (2.098) | | 0.270 (1.612) | |

Note:

Δ: Difference in admission number for a patient group between pre- and post- article release one-week periods

ΔFemale: Δ for female patients, ΔMale: Δ for male patients, ΔYounger: Δ for younger (≤ 35 years) patients, ΔOlder: Δ for older (> 35 years) patients

[a]Δ for the patient group by Female decedent included 0, and Younger decedent included 0 was used as reference for comparison with Wilcoxon rank sum tests.

Table 7 presents Δs for patient groups classified by reported suicide method. Some suicide methods showed significant differences in Δs for patient groups compared to the reference, although their effect sizes were less than moderate the following part. Specifically, ΔFemale differences were greater for Gas poisoning and Other methods, ΔMale differences were smaller for Other methods, and ΔYounger differences were smaller for Firearm discharge but greater for Other methods, compared to the reference group of No method reported.

The effect size r for the suicide methods with a significant difference ($p < 0.025$) compared to the reference were as follows: Firearm discharge for ΔYounger, −0.140; Gas poisoning for ΔFemale and ΔMale, 0.152 and −0.238, respectively; and Other methods for ΔFemale and ΔYounger, 0.240 and 0.249, respectively.

## Discussion

This study aimed to explore the associations between suicide news reports and the differences in admission numbers for sex and age SA patient groups during the one-week periods before and after the news release. Before delving into the main issue of differential susceptibility to the characteristics of suicide news among these groups, this report examines the characteristics of the newspaper reports and SA patients included in the research, as well as the changes in the admission numbers of these patient groups..

### Characteristics of newspaper suicide reports and SA patient admissions

The characteristics of the newspaper reports and SA patients addressed in this study are examined by comparing them with other statistics and reports from Japan. When compared with data from Japan's Suicide Prevention White Paper 2016 [36], which falls within the midpoint of the study period, the sex ratio and average age of suicide decedents in the studied newspaper articles were found to be similar—2:1 and approximately 10 years younger, respectively. Regarding suicide methods, the 4th and 5th most frequent methods in this study (Firearm discharge and Burning) were not included among the six most frequent methods listed in the White Paper. The disproportionate news coverage of suicides involving younger people and these methods likely stems from their significant social impact and eye-catching nature.

With respect to the SA patient subjects of this study, their sex ratio, average age, suicide methods, and psychiatric diagnoses do not significantly differ from those reported in Japanese ECC Centers, according to a meta-analysis of pooled data by Kawashima et al. [37]. The psychiatric diagnoses and SA methods observed were consistent with the figures reported by Kawashima et al.—F3, 30%; F4, 27%; F2, 13%; F6, 13%; and F1, 4%; Overdosing, 52%; Cutting, 18%; Jumping from a height, 12%; and Burning, 4%.

Thus, it is suggested that the newspaper reports and SA patients in this study did not markedly differ from general statistics in Japan, except for some bias influenced by the newsworthiness of the suicide reports.

**Table 6. Differences (Δs) in admission numbers for patient groups classified by incident types.**

| Articles: incident types (N) | ΔFemale Mean (SD) | p-value [a] | ΔMale Mean (SD) | p-value [a] | ΔYounger Mean (SD) | p-value [a] | ΔOlder Mean (SD) | p-value [a] |
|---|---|---|---|---|---|---|---|---|
| Famous person (108) | 0.657 (2.222) | 0.036 | 0.019 (1.374) | 0.388 | 0.380 (2.303) | 0.211 | 0.296 (1.174) | 0.526 |
| Multiple suicide (101) | 0.317 (2.186) | 0.799 | −0.099 (1.513) | 0.124 | 0.139 (1.828) | 0.814 | 0.097 (1.736) | 0.686 |
| Homicide-suicide (344) | −0.157 (2.157) | 0.760 | 0.155 (1.440) | 0.662 | 0.204 (2.032) | 0.521 | 0.108 (1.706) | 0.715 |
| Combination type (28) | −0.214 (2.115) | 0.547 | 0.571 (1.399) | 0.205 | 0.000 (2.000) | 0.904 | 0.357(1.545) | 0.466 |
| Unclassified (624) | 0.170 (2.002) | | 0.181(1.632) | | 0.133 (1.935) | | 0.218 (1.724) | |

Note:

Incident types was treated here as a single categorical variable

Δ: Difference in admission number for a patient group between pre- and post-article release one-week periods

ΔFemale: Δ for female patients, ΔMale: Δ for male patients, ΔYounger: Δ for younger (≤ 35 years) patients, ΔOlder: Δ for older (> 35 years) patients

[a] Δ for the patient group by Unclassified type was used as reference for comparison with Wilcoxon rank sum tests.

**Table 7. Differences (Δs) in admission numbers for patient groups classified by reported suicide method.**

| Articles: Suicide method | ΔFemale Mean (SD) | p-value[a] | ΔMale Mean (SD) | p-value[a] | ΔYounger Mean (SD) | p-value[a] | ΔOlder Mean (SD) | p-value[a] |
|---|---|---|---|---|---|---|---|---|
| Hanging (252) | 0.071 (2.139) | 0.767 | 0.147 (1.558) | 0.481 | 0.127 (2.010) | 0.671 | 0.091 (1.811) | 0.669 |
| Jumping height (225) | 0.413 (2.113) | 0.056 | 0.124 (1.568) | 0.123 | 0.227 (2.015) | 0.808 | 0.311 (1.706) | 0.220 |
| Railroad jumping (209) | −0.005 (1.989) | 0.771 | 0.225 (1.615) | 0.287 | −0.100 (1.882) | 0.125 | 0.321 (1.764) | 0.619 |
| Firearm discharge (123) | −0.252 (2.075) | 0.358 | 0.041 (1.417) | 0.098 | −0.309 (1.878) | 0.018 | 0.098 (1.581) | 0.726 |
| Burning (74) | 0.773 (2.092) | 0.039 | 0.027 (1.433) | 0.108 | 0.554 (2.234) | 0.472 | 0.226 (1.537) | 0.783 |
| Gas poisoning (65) | 0.800 (1.872) | 0.021 | −0.400 (1.445) | <0.001 | 0.585 (1.704) | 0.177 | −0.185 (1.467) | 0.215 |
| Cutting or stabbing (42) | 0.310 (2.170) | 0.365 | 0.238 (1.665) | 0.970 | 0.405 (1.988) | 0.419 | 0.143 (1.945) | 0.950 |
| Other Methods (47) | 1.191 (1.650) | < 0.001 | 0.383 (1.812) | 0.594 | 1.234 (2.169) | <0.001 | 0.340 (1.809) | 0.376 |
| No method reported (168) | 0.030 (2.118) | | 0.298 (1.421) | | 0.179 (1.855) | | 0.149 (1.673) | |

Note:

Suicide method was treated here as a single categorical variable

Δ: Difference in admission number for a patient group between pre- and post-article release one-week periods

ΔFemale: Δ for female patients, ΔMale: Δ for male patients, ΔYounger: Δ for younger (≤ 35 years) patients, ΔOlder: Δ for older (> 35 years) patients

[a]Δ for the patient group by No method reported was used as a reference for comparison with Wilcoxon rank sum tests.

## Associations between newspaper suicide reports and SA patient admissions

This study identified a significant increase in SA admissions during the week following the release of suicide reports across all sex and age patient groups, consistent with previous findings of SA increases following celebrity suicides. [28,38,39] The results underscore the need for continued vigilance in the style and content of newspaper reports, even after considerable efforts have been made to adhere more closely to WHO media recommendations.

The finding that reporting on younger suicide decedents was correlated with a decrease in subsequent overall admissions of SA patients suggests that such articles may have a suppressive effect by evoking negative feelings towards suicide or SA. This finding aligns with that of Sinyor et al. [23] on cases of suicide.

With regard to reported suicide methods, this study found that reports of firearm suicides were associated with a decrease in overall SA patient admissions, suggesting a suppression effect. However, this finding contrasts with previous studies that reported an increase in suicide cases following firearm suicide reports in other countries. [9,23] The primary reason for this discrepancy may lie in Japan's unique circumstances of long-standing, strict gun control policies, which have led to an exceptionally low civilian firearm possession rate of 0.3 per 100 people in 2017, compared to the global

average of 6.6. [40] Firearm suicides may be especially feared and stigmatized in Japan, potentially contributing to a suppression of SA incidents following the news release.

Reporting of Infrequently reported suicide methods was also associated with increased SA patient admissions. This association can be attributed to the dramatic or violent nature of these methods such as self-explosion. This observation is in line with Thomas et al.'s contention [41] that attention-grabbing suicide methods, especially when used by celebrities, have a strong potential to instigate imitative suicides.

It was recognized that reporting on other suicide methods was not associated with changes in the overall number of admissions. By contrast, in previous reviews on suicide cases, [1,5,7,11,13,15,27] have reported that media coverage of specific suicide methods influenced the occurrence of suicides. Some studies have found that certain methods, such as jumping from a height, [8,14,23] are associated with an increase in suicide incidents, whereas others, such as cutting or stabbing, and railroad jumping, [23] may be associated with a decrease in suicide rates.

This discrepancy may be explained by differences in the research focus—specifically, whether the outcome examined was suicide or suicide attempt—and by variations in the socio-cultural backgrounds of the study settings.

## Associations between sex and age-related article characteristics of suicide decedents and sex and age group admissions

It has been reported that the sex and age of reported suicide decedents are often associated with the sex and age of subsequent suicides following such reports. [14,24–26,42–44] However, in the present study, nonparametric MANOVA did not find any significant associations between the sex and age-related article characteristics of suicide decedents and the admissions of sex and age groups of SA patients.

Regarding the sex and age groups of suicide decedents, this study's findings were limited to a significant decrease in overall SA patient admissions and a trend-level decrease in female patient admissions following the release of reports on younger suicide decedents. These findings are unlikely to be attributed to identification or social learning processes related to SA. Instead, they could be interpreted as a psychological response to the deaths of younger individuals in suicide incidents, potentially mediated by negative emotions towards suicide itself.

One reason for this discrepancy could be the difference in the types of suicide incidents studied: this study focused on routine suicide reports, which generally have less impact compared with the celebrity suicides examined in previous studies. [11,27] Additionally, the implementation of WHO media recommendations in Japan since the early 2000s [6] may have contributed to this difference. The recommendations have been progressively adopted by journalists to mitigate the risk of instigating strong responses such as identification.

## Associations between types of suicide incidents and sex and age group admissions

Regarding suicide incident types, celebrity suicides are typically associated with subsequent occurrences of suicide [11,25,27] and SAs [38,39]. However, this study did not find a significant association between famous person suicides, which include celebrity suicides, and increased admission numbers across sex and age groups. Only a trend-level increase in female patient admissions following article release was observed. The nonparametric MANOVA to test this association and yielded a non-significant result. It is important to note that the "famous person suicides" examined in this study covered a wider range of incidents, including those with less impact compared with "celebrity suicides." This aligns with the observation that fewer studies have reported post-article release increases in suicide incidents following non-celebrity suicides [11,27].

The negative associations observed in previous studies between suicide incident reports with negative connotations, such as homicide-suicides and multiple suicides, and decreases in the number of suicide incidents [23,25,45] were not observed in the present study. One possible explanation is that non-sensational and restrained reporting, in accordance with media recommendations, significantly reduced the impact of these articles. Additionally, the overlap between these

two types of suicide incident and the suicide referred to as Shinju, the latter of which is regarded with relatively lenient attitudes in Japan, may partly explain the lack of changes in SA occurrences following reports of these suicides in this study. In Western countries, it has been reported that the majority of individuals committing homicide-suicide are male.[46,47] However, in Japan, Shinju often occurs within mother-child relationships.[32] The finding in this study that there was no significant difference in sex among the perpetrators of homicide-suicide suggests that many of these cases involved mother-child Shinju. Likewise, the observation that the majority of the reported multiple suicide cases involved female decedents also indicates that many of these incidents may have occurred within intra-family relationships.

## Associations between reported suicide methods and sex and age group admissions

The finding that the reporting of suicide methods was associated with differences in admission numbers across sex and age groups of SA patients suggests that reporting on specific suicide methods influenced admission numbers in particular patient groups. For example, the analysis identified that the younger patient group showed a negative association between reports of firearm suicides and their admission number. This finding helps to explain the correlation between the reported suicide methods and changes in the overall admission number observed in the preliminary correlation analysis.

Reports of gas poisoning, mostly involving charcoal burning, showed bidirectional associations: an increase in female admissions and a decrease in male admissions. These opposing findings may reflect sex differences in psychological responses to gas poisoning. This method was more frequently used by females and in group suicide incidents, as observed in this study and in the Japan Suicide Prevention White Paper. [36] Furthermore, the charcoal burning method has been reported to have a strong modeling effect that elevates suicide rates, [39,41] and it is often chosen by those who express a desire for a peaceful death [48]. These aspects of gas poisoning may help explain the differing psychological responses between the sexes in this study.

The associations of increased admissions for female and younger patient groups with reports on infrequently reported suicide methods suggest that these specific patient groups are key contributors to the significant correlations observed between the methods and the overall increase in patient admissions following article release in this study.

## Implications of associations between suicide news reports and subsequent SA patient admissions

The associations found in this study may indicate an interaction between the characteristics of suicide incident articles and the respective recipient populations. Although identification and social learning theories provide a framework to bridge these two aspects, this study did not find a correspondence between reported suicide decedents and suicidal patients in terms of sex and age group, which has been argued as an indication of identification in previous studies. The lack of evidence supporting the identification theory in this study may be due to its focus on routine (low impact) suicide reports, as well as the implementation of media recommendations that emphasize reducing detailed descriptions of suicide decedents to avoid identification.

Instead, the results of this study underscore the differential influences of reporting of some suicide methods on suicidal patients across sex and age groups. The differences are likely better explained by psychological responses on the part of the recipients, as indicated by the opposing trends in admission numbers between both sexes following the release of articles involving gas poisoning, and by increased admissions among female and younger patient groups following the release of articles on infrequently reported suicide methods.

It is also important to emphasize the role of socio-cultural factors in the respective regions in interpreting the results of this study. A notable example is the decrease in the number of younger patient admissions following the release of articles on firearm suicide. This response is likely influenced by the extremely strict gun control policies in Japan. Thus, these differential associations are likely to arise from the psychological responses and socio-cultural conditions across sex and age groups of suicidal individuals.

## Limitations

This study has several limitations that should be acknowledged. First, it utilized articles from the four largest newspapers in Japan as the source of media suicide reports, while other news sources, such as television stations and internet media, were not included. The suicide reports from other media may differ in nature from those of newspapers, particularly in terms of adherence to the WHO media recommendations. Second, selecting only initial news articles for analysis to achieve a simplified data structure introduces some problems, including the inability to assess the cumulative effects of follow-up reports. Future research should aim to incorporate and analyze other types of coverage. Third, this study was conducted at only one ECC center in a region of the Tokyo metropolitan area, which may limit the generalizability of the findings to populations of other areas. Fourth, there are limitations associated with the use of medical record review, including the inability to eliminate selection bias in identifying SA patients, to ensure the validity and reliability of assessments, and the lack of direct access to patients' experiences of receiving media messages. Further research should employ standardized assessments for suicide attempts and conduct interview-based surveys, as in previous studies. [38,39,48] Fifth, some analyses may have been affected by limited statistical power. For example, fine categorization reduced group sizes, increasing the risk of type II errors, and broad age groupings of reported suicide decedents and SA patients constrained the clarity of conclusions. Lastly, most of the previous studies referenced in the discussion, were primarily focused on suicide, due to the relative lack of research on SA. Since the relationship between media reports and the occurrence of suicide is not identical to that of SA, it is necessary to consider these differences when interpreting the results.

## Conclusions

This study investigated the relationship between the basic characteristics of newspaper suicide reports and changes in the number of admissions for SA patients at an ECC center in Tokyo over approximately 7 years. The findings support previous studies that posited the influence of suicide newspaper reports on occurrences of SA. Notably, the influence of media activities on SA occurrence was observed even under conditions where the articles were routine, low impact, and mostly adhered to WHO recommendations. This underscores the importance of continued efforts to diminish and ultimately nullify the harmful effects of media suicide reporting.

The nonparametric MANOVA employed in this study elicited a significant association between the reporting on suicide methods and the number of admissions across sex and age groups of SA patients. Suicide method reporting of gas poisoning, firearm discharge, and infrequent methods was associated with change in admission numbers for certain SA patient groups. These associations are likely to be interpretable within psychological and socio-cultural contexts. On the other hand, the associations with sex- and age-related article characteristics and types of suicide incidents were not significant.

In this study, the lack of significant findings supporting the identification theory—such as the lack of correspondence between the sex and age of reported suicide decedents and those of SA patients, as well as the non-significant impact of famous person suicides—suggests that, in the context of routine suicide reports and the widespread adoption of media recommendations, susceptibility is more strongly influenced by individual psychological and regional sociocultural factors of news recipients, rather than by the characteristics of the reported suicide decedents.

The differential association patterns between suicide method reporting and subsequent changes in admission numbers provide valuable insights into the susceptibility of suicidal individuals to media coverage, highlighting broader implications for various sex and age groups, with both harmful and protective effects. Moreover, they highlight the importance of tailoring psychoeducational interventions to the unique response profiles of different populations. Building on these findings, future research should explore the factors underlying sex- and age-related differences in media responsiveness across diverse sociocultural contexts, with the goal of optimizing suicide prevention programs and treatment strategies for SA patients.

## Supporting information

**S1. Newspaper article data.**
(XLSX)

**S2. Newspaper incident data.**
(XLSX)

**S3. Newspaper suicide decedent data.**
(XLSX)

**S4. Admitted SA patient data.**
(XLSX)

**S5. Pre- and post-article release admission numbers.**
(XLSX)

## Author contributions

**Conceptualization:** Naoki Hayashi.

**Investigation:** Yasushi Emura, Sho Kanata, Naoki Hayashi, Ken-ichi Matsumura, Akihisa Akahane.

**Methodology:** Naoki Hayashi.

**Resources:** Ken-ichi Matsumura, Akihisa Akahane.

**Validation:** Yasushi Emura, Sho Kanata.

**Writing – original draft:** Naoki Hayashi.

**Writing – review & editing:** Yasushi Emura, Sho Kanata, Naoki Hayashi, Mamoru Tochigi, Hiroshi Kunugi.

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
