## [Decision Letter · Decision Letter 0]

11 Feb 2025

PONE-D-24-46119Sex and age differences in the association between routine suicide newspaper reporting and change in admissions of suicidal patients: An investigation at an emergency and critical care center in TokyoPLOS ONE

Dear Dr. Hayashi,

Thank you for submitting your manuscript to PLOS ONE. After careful consideration, we feel that it has merit but does not fully meet PLOS ONE’s publication criteria as it currently stands. Therefore, we invite you to submit a revised version of the manuscript that addresses the points raised during the review process.

We look forward to receiving your revised manuscript.

Kind regards,

Paolo Roma

Academic Editor

PLOS ONE

Journal Requirements:

Reviewers' comments:

Reviewer's Responses to Questions

**Comments to the Author**

1. Is the manuscript technically sound, and do the data support the conclusions?

Reviewer #1: Yes

Reviewer #2: Yes

Reviewer #3: Yes

2. Has the statistical analysis been performed appropriately and rigorously? 

Reviewer #1: Yes

Reviewer #2: Yes

Reviewer #3: I Don't Know

3. Have the authors made all data underlying the findings in their manuscript fully available?

Reviewer #1: Yes

Reviewer #2: Yes

Reviewer #3: Yes

4. Is the manuscript presented in an intelligible fashion and written in standard English?

Reviewer #1: Yes

Reviewer #2: Yes

Reviewer #3: Yes

5. Review Comments to the Author

Reviewer #1: Thank you for the opportunity to review this manuscript. My only question about the context of this study was whether young people in Japan still read newspapers or get their news from other media outlets such as social media and how that could be more of an impact/ be a confounding factor especially because newspapers as you have noted do follow WHO recommendations, but other sources of information might not.

Reviewer #2: The study design is comprehensive, but it would be helpful to include a more detailed explanation of how the specific suicide methods reported in the newspapers were categorized and analyzed. This would provide clearer insight into how these methods were considered in the context of their impact on different populations.

While the statistical analyses are robust, further elaboration on the choice of nonparametric tests and why they were preferred over parametric tests for this dataset would enhance the transparency of the methodological approach.

The authors do an excellent job of linking their findings to psychological and sociocultural factors, but they might want to expand on the implications for future suicide prevention programs, especially considering the differential effects observed across sex and age groups.

In some parts of the manuscript, particularly in the methods section, there is heavy reliance on jargon and complex terminology that could be simplified for broader audiences, including policymakers and mental health practitioners.

Reviewer #3: Thank you for the opportunity to review this manuscript titled "Sex and Age Differences in the Association between Routine Suicide Newspaper Reporting and Change in Admissions of Suicidal Patients: An Investigation at an Emergency and Critical Care Center in Tokyo." The study addresses an important topic related to media influence on suicidal behavior and its differential impact across sex and age groups. Strengths of the study include the use of an extensive dataset spanning multiple years and the application of statistical methods to analyze associations between media reporting and emergency admissions. However, there are methodological and statistical limitations that should be addressed to improve the robustness and clarity of the findings.

In general, the Authors may benefit from professional editing to improve grammatical consistency and enhance the manuscript’s overall flow.

Introduction.

This section provides a comprehensive background and follows a logical progression.

Methods.

The Authors should clarify why specific variables (age, sex) were prioritized over others, as their relevance to broader outcomes is not immediately evident.

Certain aspects of this section lack clarity. For example, the reference to “search terms” raises questions about whether the search was conducted using physical or digital copies of newspapers.

The operational definitions of suicide attempts and their severity require further elaboration. Were standardized assessment tools used to classify suicidality beyond the items mentioned in lines 167-168?

The study refers to a Bonferroni correction (line 199), but it is unclear whether this adjustment was consistently applied across all analyses. Were adjustments made for multiple comparisons to control for Type I errors?

Multiple/murder-suicides represent distinct phenomena. The Authors should provide a stronger rationale for their inclusion in the study.

Lines 117-118. The Authors should provide a rationale for excluding follow-up news reports. Moreover, in cases where the same event was reported by multiple sources, only the first report was considered. However, repeated exposure to an event through multiple sources could be an important factor. Did the Authors account for the potential influence of cumulative reporting on admissions?

The terms "article-based data," "incident-based data," and "suicide-decedent-based data" require clearer definitions. Additionally, the structure of the first part of the results, which describes and analyses characteristics of suicides and suicide attempts, should be reorganized to align with these distinctions.

Statistical Analyses. A more detailed justification of the statistical methods is needed. The rationale and objectives behind these analyses should be clearly articulated. Additionally, effect sizes should be reported where applicable.

Table 1. The rationale and purpose of the chi-square analyses should be clarified. The placement of another chi-square analysis immediately after the table may be confusing to readers and should be reconsidered without an explanation. Furthermore, some sample sizes in these analyses are quite small, which should be acknowledged as a limitation. In general, overly complex tables may be difficult to interpret without extensive cross-referencing.

Results.

Lines 432, 525. The terms "effect" and "impact" imply causation, which cannot be inferred from the current study. More cautious language should be used.

The study reports a suppression effect for firearm-related suicides, which is surprising especially given the inclusion of murder-suicides. The explanation attributing this finding to strict gun control is not entirely convincing, as firearms have been associated with violent behaviors even in countries with stringent regulations (e.g., Colasanti et al., 2024, on homicide-suicide in Italy). Further discussion of this finding is warranted.

Discussion.

The non-significant findings regarding the sex and age characteristics of reported suicide decedents are unexpected given prior research. The Authors should explore potential explanations for these results in greater depth.

The weak but significant correlations between Δs and article characteristics (e.g., suicide method) are informative but require deeper interpretation. The Authors should discuss why certain methods show stronger correlations and how these findings align with existing literature.

Limitations.

Studies that rely on newspaper reports have inherent limitations, including biases in reporting—such as the overrepresentation of dramatic or violent suicide methods. While the Authors briefly acknowledge this, a more detailed discussion of these biases is needed. Similarly, the potential impact of the study’s regional focus should be addressed.

Additional Suggestions.

The authors might consider including a figure to visually depict trends in suicide attempt admissions over time, potentially disaggregated by patient group.

Although it may be beyond the scope of this study, future research could explore seasonal variations and conduct time-series analyses.

6. PLOS authors have the option to publish the peer review history of their article (what does this mean? ). If published, this will include your full peer review and any attached files.

**Do you want your identity to be public for this peer review?** For information about this choice, including consent withdrawal, please see our Privacy Policy .

Reviewer #1: No

Reviewer #2: **Yes: ** Prof Dr Saad Alatrany

Reviewer #3: No

---

## [Author Response · Author response to Decision Letter 1]

4 Apr 2025

Our responses to comments by reviewers:

Reviewer #1:

Comment 1-1

My only question about the context of this study was whether young people in Japan still read newspapers or get their news from other media outlets such as social media and how that could be more of an impact/ be a confounding factor especially because newspapers as you have noted do follow WHO recommendations, but other sources of information might not.

Response 1-1

The subscription rate of newspapers is an important indicator of their influence. In Japan, subscription rates have been gradually declining, particularly among younger generations, making it necessary to include these figures in this paper. According to the report titled 2015 National Media Contact and Evaluation Survey Report conducted by The Japan Newspaper Publishers & Editors Association as part of the National Survey on Media Usage and Evaluation, the overall subscription rate in 2015, which was approximately the midpoint of the study period (April 2012 to January 2019), was 77.7%. Meanwhile, the subscription rate among those in their 30s, close to the median age group of admitted patients in this study, was 40.5%. Subscription rates across all generations have been declining year by year. For instance, in 2021, the overall subscription rate and the subscription rate among those in their 30s were 60.9% and 32.4%, respectively. Some of these figures are incorporated into the main text of the paper.

In 2015, approximately the midpoint of the study period, the newspaper subscription rate in Japan was reported to be 77.7% across all age groups and 40.5% among those in their 30s. (31) (Lines 113-115)

As noted by the reviewer, SA (suicide attempt) occurrence is thought to be influenced by the level of adherence to WHO recommendations. However, in this study, the adherence status of media other than newspapers has not been clarified, necessitating some reservation in interpreting the results. This has been added to the limitations as a potential complicating factor.

The suicide reports from other media may differ in nature from those of newspapers, particularly in terms of adherence to the WHO media recommendations. (Lines 605-607 in Limitations)

Reviewer #2:

Comment 2-1

It would be helpful to include a more detailed explanation of how the specific suicide methods reported in the newspapers were categorized and analyzed. This would provide clearer insight into how these methods were considered in the context of their impact on different populations.

Response 2-1

SA methods selected for analysis included the most frequent seven suicide methods among the 771 reported suicide decedents. The sentence below is inserted in Revised manuscript.

Assessed suicide methods included Hanging, Railroad jumping, Jumping from a height (Height jumping), Gas poisoning, Firearm discharge, Burning, and Cutting or stabbing. For analysis, the items used were the most frequently observed methods among reported suicide decedents, Other methods, and No method reported. (Lines 138-142)

Comment 2-2

While the statistical analyses are robust, further elaboration on the choice of nonparametric tests and why they were preferred over parametric tests for this dataset would enhance the transparency of the methodological approach.

Response 2-2

MANOVA is an appropriate analytical method for examining the relationships between interrelated dependent variables and independent variables. Furthermore, in this study, since the matrix of dependent variables is singular (The sum of ΔFemale and ΔMale equals the sum of ΔYounger and ΔOlder.), the nonparametric MANOVA was required for the analysis. The following sentence has been added to the manuscript:

The nonparametric MANOVA model was crucial for evaluating the associations while accounting for the interrelationships among the dependent variables. More importantly, it resolved the issue of matrix singularity among these variables in the analysis. (Lines 203-206)

Comment 2-3

The authors do an excellent job of linking their findings to psychological and sociocultural factors, but they might want to expand on the implications for future suicide prevention programs, especially considering the differential effects observed across sex and age groups.

Response 2-3

In response to this comment, we have organized and presented a discussion on the possible differential impacts for each sex and age group population, which represents a key finding of this study. In the revised manuscript, these findings of this study and their implications have been discussed in relation to the treatment of SA patients and suicide prevention.

We have added the following statements to the conclusion section to support these points.

In this study, the lack of significant findings supporting the identification theory—such as the absence of correspondence between the sex and age of reported suicide decedents and those of SA patients, as well as the non-significant impact of famous person suicides—suggests that, in the context of routine suicide reports and the widespread adoption of media recommendations, susceptibility is more strongly influenced by individual psychological and regional sociocultural factors of news recipients, rather than by the characteristics of the reported suicide decedents.(Lines 636-642)

Moreover, they (the findings) highlight the importance of tailoring psychoeducational interventions to the unique response profiles of different populations. Building on these findings, future research should explore the factors underlying sex- and age-related differences in media responsiveness across diverse sociocultural contexts, with the goal of optimizing suicide prevention programs and treatment strategies for SA patients. (Lines 646-650)

Comment 2-4

In some parts of the manuscript, particularly in Methods section, there is heavy reliance on jargon and complex terminology that could be simplified for broader audiences, including policymakers and mental health practitioners.

Response 2-4

To ensure a smoother description of the suicide situation in Japan while avoiding the use of "jargon and complex terminology," we revised the terms and added explanations where necessary. First, we replaced the term "murder-suicide" with "homicide-suicide," which has become a more commonly studied research topic in recent years. Although the term "multiple suicide" is somewhat ambiguous and may have varied interpretations, we continued to use it, defining it as incidents involving multiple suicide decedents. Furthermore, we have added an explanation for the term "Shinju," describing it as a homicide-suicide or multiple suicide occurring within particularly intimate relationships. We also included a cultural note explaining that, in Japan, "Shinju" has historically been a theme in theater and literature, contributing to a cultural tendency to empathize with such acts.

…“Jisatsu (suicide)” and “Shinju (homicide-suicide or multiple suicide that occurred within particularly intimate relationships).” "Shinju" is a term unique to Japan and has long been a theme in Japanese theater, literature, and other cultural expressions.(32) (Line 120-122)

Incident characteristics examined included homicide-suicide (cases involving the murder of another person prior to the suicide) and multiple-suicide (cases involving more than one decedent, including serial or group suicides) as prominent features of suicide. (Lines 128-130)

Further changes regarding 'Shinju' will be described in Response 3-15.

Reviewer #3:

Comment 3-1

The Authors should clarify in Method section why specific variables (age, sex) were prioritized over others, as their relevance to broader outcomes is not immediately evident.

Response 3-1

As indicated in the Introduction section, sex and age have been central to previous research. Specifically, the identification theory highlights the correspondence between the sex and age groups of suicide decedents and media recipients as a key issue. To emphasize the importance of these variables, we have added the highlighted phrase below to the list of investigation items in Methods section.

Furthermore, we have organized the order of the investigated items to maintain consistency, and the explanations of the items have been presented in that order.

To investigate article characteristics, we chose items of the sex and age of suicide decedents, which are central to the focus of this study, their social status, occupation, and the suicide methods used. Incident characteristics examined included homicide-suicide (cases involving the murder of another person prior to the suicide) and multiple-suicide (cases involving more than one decedent, including serial or group suicides) as prominent features of suicide. (Lines 126-130)

Additionally, as indicated in Response 2-3, we have enhanced the descriptions in the Conclusion subsection to link the findings of this study more explicitly to the main focus on sex and age group differences, ensuring consistency throughout the manuscript.

Comment 3-2

Certain aspects of Method section lack clarity. For example, the reference to “search terms” raises questions about whether the search was conducted using physical or digital copies of newspapers.

Response 3-2

The search was conducted using digital copies of the newspapers. The term "Jisatsu (suicide)" and “Shinju” were searched across the full text of articles in a news database. This has been explicitly clarified in the Methods section as follows.

A digital search was conducted across the full text of articles using the search terms “Jisatsu (suicide)” and “Shinju (homicide-suicide or multiple suicide that occurred within particularly intimate relationships). (Lines 119-121)

The 'lack of clarity' is presumed to stem from the use of 'Shinju' as a search term, as well as the inconsistent order in presenting and explaining the investigation items. Please refer to Response 2-4 for our actions addressing these issues.

Comment 3-3

The operational definitions of suicide attempts and their severity require further elaboration. Were standardized assessment tools used to classify suicidality beyond the items mentioned in lines 167-168 (beyond the description for items 246 and 250 of SADS)

Response 3-3

Our Consultation Liaison team, who provided clinical DATA of SA patients for the study, did not use standardized assessment tools. Instead, we have added definition of SA eligible for treatment by our hospital’s Consultation Liaison team.

SA patients were individuals hospitalized for the treatment of physical injuries resulting from suicide attempts, which are defined as concrete actions based on suicidal ideation,(33) …(Line 165)

Comment 3-4

The study refers to a Bonferroni correction (in Methods section), but it is unclear whether this adjustment was consistently applied across all analyses. Were adjustments made for multiple comparisons to control for Type I errors?

Response 3-4

Since each suicide incident was reported by two newspapers on average, we applied the Bonferroni correction, making the significance level twice as stringent, based on the rationale that approximately two tests were conducted per incident, and that the analysis aimed to assess the effect of a single article per incident. In contrast, for tests involving sex differences in the data of suicide decedents or suicide incidents, in which duplicate cases were excluded, and therefore, this correction was deemed unnecessary.

The explanation in Response 3-7 would contribute to understanding the use of the Bonferroni correction in handling the article data.

For the analyses of news articles, since each incident was covered slightly less than twice on average, the significance level was set at 0.025 (0.05/2) (two-tailed) using Bonferroni's correction. For the analyses of data with duplication-eliminated incident cases, a significance level of 0.05 was applied. (Lines 212-215)

Comment 3-5

Multiple/murder-suicides represent distinct phenomena. The Authors should provide a stronger rationale for their inclusion in the study to make this situation clear.

Response 3-5

Homicide-suicide and multiple suicide were selected as prominent features of suicide incidents. We have added the following description to clarify this point.

Incident characteristics examined included homicide-suicide (cases involving the murder of another person prior to the suicide) and multiple-suicide (cases involving more than one decedent, including serial or group suicides) as prominent features of suicide. (Line 130)

Comment 3-6

The Authors should provide a rationale for excluding follow-up news reports. Moreover, in cases where the same event was reported by multiple sources, only the first report was considered. However, repeated exposure to an event through multiple sources could be an important factor. Did the Authors account for the potential influence of cumulative reporting on admissions?

Response 3-6

This study focused on the short-term impact within one week after an incident. It was limited to initial reports issued within one week following suicide incidents, and the subjects of the study were SA patients who presented within one week after the reports were released. Follow-up articles were excluded, as they are often published some time after the incident and do not reflect the immediate impact of the incident.

The following descriptions have been added in the manuscript.

This study examined the association between the characteristics of initial suicide reports in four major newspapers and the differences in admission numbers (Δs) of SA patients across sex and age groups during one-week pre- and post-article release periods. (Line 30)

The aim of this study was to explore the relationship between newspaper initial reports of suicide incidents and hospital admissions of SA patients immediately following the release of the report. (Lines 95-96)

To examine the association between the characteristics of initial suicide reports immediately after incidents and the short-term change in SA patient admissions, we selected articles reporting actual suicides within a week of publication, excluding follow-up reports. (Lines 123-125)

Comment 3-7

The terms "article-based data," "incident-based data," and "suicide-decedent-based data" require clearer definitions. Additionally, the structure of the first part of the results, which describes and analyses characteristics of suicides and suicide attempts, should be reorganized to align with these distinctions.

Response 3-7

Incident-based data' and 'suicide-decedent-based data' were used in the preliminary analysis of this study, with the primary analysis focusing on 'article-based data.' 'Incident-based data' was created to present the characteristics of reported incidents while eliminating duplicated incident reports. 'Suicide-decedent-based data' consists of information on all suicide decedents from the duplication-eliminated incident data. All of these data were utilized in the analysis and are also included in the supporting files.

In Discussion section's 'Characteristics of Newspaper Suicide Reports and SA Patient Admissions' subsection, these data were used to elucidate the features of the reports examined in this study by comparing them with other suicide and SA statistics.

In the revised manuscript, we have opted not to use the terms 'incident-based data,' 'suicide-decedent-based data' and 'article-based data.’ Instead, we have provided explanations in Methods section and in the Notes for Tables 1 and 2 as follows.

To avoid duplication when presenting the characteristics of suicide decedents and incidents reported in multiple news articles, data from the earliest article were chosen. Furthermore, if multiple reports covering the same incident were published on the same day, data from the newspaper with the largest circulation were used. (Lines 143-146)

[Restatement] For the analyses of news articles, since each incident was covered slightly less than twice on average, the significance level was set at 0.025 (0.05/2) (two-tailed) using Bonferroni's correction. For the analyses of data with duplication-eliminated incident cases, a significance level of 0.

---

## [Decision Letter · Decision Letter 1]

21 May 2025

PONE-D-24-46119R1Sex and Age Differences in the Association between Routine Suicide Newspaper Reporting and Change in Admissions of Suicidal Patients: An Investigation at an Emergency and Critical Care Center in TokyoPLOS ONE

Dear Dr. Hayashi,

Thank you for submitting your manuscript to PLOS ONE. After careful consideration, we feel that it has merit but does not fully meet PLOS ONE’s publication criteria as it currently stands. Therefore, we invite you to submit a revised version of the manuscript that addresses the points raised during the review process.

We look forward to receiving your revised manuscript.

Kind regards,

Paolo Roma

Academic Editor

PLOS ONE

Journal Requirements:

Reviewers' comments:

Reviewer's Responses to Questions

**Comments to the Author**

1. If the authors have adequately addressed your comments raised in a previous round of review and you feel that this manuscript is now acceptable for publication, you may indicate that here to bypass the “Comments to the Author” section, enter your conflict of interest statement in the “Confidential to Editor” section, and submit your "Accept" recommendation.

Reviewer #3: (No Response)

Reviewer #4: All comments have been addressed

2. Is the manuscript technically sound, and do the data support the conclusions?

Reviewer #3: Yes

Reviewer #4: Yes

3. Has the statistical analysis been performed appropriately and rigorously? 

Reviewer #3: Yes

Reviewer #4: Yes

4. Have the authors made all data underlying the findings in their manuscript fully available?

Reviewer #3: Yes

Reviewer #4: Yes

5. Is the manuscript presented in an intelligible fashion and written in standard English?

Reviewer #3: Yes

Reviewer #4: Yes

6. Review Comments to the Author

Reviewer #3: The Authors have made substantial and generally successful efforts to address my concerns. Most of the original comments have been responded to with care and, in several cases, led to meaningful improvement in the manuscript. However, some areas still require further attention before the manuscript is suitable for publication.

1. The operational definition of suicide attempts relies on non-standardized, clinician-rated criteria without the use of validated instruments. While the Authors clarify that SADS items were referenced and provide a working definition of SA, the lack of structured assessments undermines the comparability and psychometric rigor of the patient data. This limitation should be explicitly acknowledged in the Methods and Limitations sections, with a note on the potential impact on validity and inter-rater reliability. Furthermore, the provided definition of SA might exclude low-lethality but high-intent attempts, potentially introducing sampling bias.

2. It appears that the Bonferroni correction was applied specifically to analyses based on article-level data, where duplication across reports was retained. At the same time, other sections indicate that duplicates were removed to avoid inflation of incident counts. If this distinction underlies the Authors’ approach, it should be made more explicit. As it stands, the rationale for correction is not clearly aligned with the data handling procedures described elsewhere. I encourage the Authors to review this carefully for consistency and clarify accordingly. In other words, the manuscript toggles between article-level data (where each article is a unit) and incident-level data (where duplicates are removed). The concern is that this distinction is not clearly mapped onto the analysis flow, leaving the reader uncertain about which dataset structure was used for which test, and whether correction thresholds were consistently applied.

3. SA patients were divided into "Younger" and "Older" groups based on whether they were below or above the median age of suicide decedents. While this choice is theoretically justifiable, especially if guided by identification theory, which emphasizes alignment between media representations (decedents) and recipients (patients), the rationale is not clearly explained in the current manuscript. I recommend that the Authors add a sentence explicitly clarifying that this cutoff was chosen not to reflect the patient population distribution, but rather to test age-related susceptibility to suicide reporting effects. In other words, the Authors should clarify this as a theory-driven choice.

4. The Authors state that follow-up news reports were excluded in order to focus on the short-term impact of initial suicide coverage within a one-week period. While this decision aligns with their analytic design, it does not fully address the concern about cumulative exposure. A growing body of literature suggest that repeated exposure to suicide-related content (over time or by multiple sources) can compound risk, particularly among vulnerable individuals. By excluding follow-up reports and analysing only the first article per incident, the Authors may inadvertently underrepresent the real-world exposure profile of individuals who consume multiple media reports on the same suicide event. While I appreciate the clarity of the short-term focus, the ecological validity of the findings is limited. The Authors should consider explicitly acknowledging this limitation or potential future direction in the manuscript, as it would strengthen the transparency of the study’s conclusions.

Reviewer #4: Sex and Age Differences in the Association between Routine Suicide Newspaper

Reporting and Change in Admissions of Suicidal Patients: An Investigation at an

Emergency and Critical Care Center in Tokyo

Thanks

7. PLOS authors have the option to publish the peer review history of their article (what does this mean? ). If published, this will include your full peer review and any attached files.

**Do you want your identity to be public for this peer review?** For information about this choice, including consent withdrawal, please see our Privacy Policy .

Reviewer #3: No

Reviewer #4: No

---

## [Author Response · Author response to Decision Letter 2]

12 Jun 2025

Our responses to comments by reviewers:

Reviewer #3:

Comment 3-1

The operational definition of suicide attempts relies on non-standardized, clinician-rated criteria without the use of validated instruments. While the Authors clarify that SADS items were referenced and provide a working definition of SA, the lack of structured assessments undermines the comparability and psychometric rigor of the patient data. This limitation should be explicitly acknowledged in the Methods and Limitations sections, with a note on the potential impact on validity and inter-rater reliability. Furthermore, the provided definition of SA might exclude low-lethality but high-intent attempts, potentially introducing sampling bias.

Response 3-1

As the study did not employ standardized assessments for suicide attempts, there is a possibility—just as the reviewer points out—that patients with more severe physical damage due to their suicide attempts were more likely to be included, introducing a potential sampling bias. It is essential to clarify that the findings of this study are based on a patient population selected under such methodological limitations. Therefore, we have made the following corrections.

Response 3-1-1

The following correction has been made

Previous manuscript

Data from an Emergency and Critical Care Center from 2012 to 2019 were used. (Lines 32-33 in Abstract)

Revised manuscript

Data from an Emergency and Critical Care Center from 2012 to 2019 were obtained through a review of medical records. (Lines 32-33 in Abstract)

Response 3-1-2

The following correction has been made

Previous manuscript

Data on SA patients admitted to the ECC Center from April 1, 2012, to January 31, 2019, were obtained by reviewing medical records (Lines 171-172)

Revised manuscript

Data on SA patients admitted to the ECC Center from April 1, 2012, to January 31, 2019, were obtained through a retrospective review of medical records. (Lines 170-171)

Response 3-1-3

The following sentence has been inserted in Limitations section.

Fourth, there are limitations associated with the use of medical record review, including the inability to eliminate selection bias in identifying SA patients, to ensure the validity and reliability of assessments, and the lack of direct access to patients’ experiences of receiving media messages. Further research should employ standardized assessments for suicide attempts and conduct interview-based surveys, as in previous studies (38, 39, 48). (Lines 616-621)

Comment 3-2

It appears that the Bonferroni correction was applied specifically to analyses based on article-level data, where duplication across reports was retained. At the same time, other sections indicate that duplicates were removed to avoid inflation of incident counts. If this distinction underlies the Authors’ approach, it should be made more explicit. As it stands, the rationale for correction is not clearly aligned with the data handling procedures described elsewhere. I encourage the Authors to review this carefully for consistency and clarify accordingly. In other words, the manuscript toggles between article-level data (where each article is a unit) and incident-level data (where duplicates are removed). The concern is that this distinction is not clearly mapped onto the analysis flow, leaving the reader uncertain about which dataset structure was used for which test, and whether correction thresholds were consistently applied.

Response 3-2

While article-level data served as the primary focus of analysis in this study, we also analyzed datasets in which duplications of suicide decedents and reported incidents across articles were removed. This was necessary to describe the characteristics of suicide decedents and incidents, and to allow comparisons with respective statistics of other kind. Because creating these duplication-removed datasets involved somewhat complex procedures, we explained the process in the previous manuscript following the description of newspaper variables and in the notes of Table 1 and Table 2.

For analyses using these duplication-removed datasets, no repeated statistical testing was conducted. However, in the analyses investigating the association between article-level data and SA patient admissions, we applied Bonferroni correction under the rationale that each incident was reported in approximately two articles, effectively leading to repeated tests per incident.

To ensure consistency and clarity in how these datasets were handled, we have made the following revisions to the manuscript.

Response 3-2-1

Concerning the use of the Bonferroni correction, additional wording has been added to clarify the rationale.

For the analyses of news articles, since each incident was covered in slightly fewer than two articles on average—effectively resulting in repeated testing—the significance level was set at 0.025 (0.05/2) (two-tailed) using Bonferroni’s correction. (Lines 209-211)

Response 3-2-2

We have expanded the explanation of the purpose behind creating the dataset with duplicate reports removed.

Previous manuscript

To avoid duplication when presenting the characteristics of suicide decedents and incidents reported in multiple news articles, data from the earliest article were chosen. Furthermore, if multiple reports covering the same incident were published on the same day, data from the newspaper with the largest circulation were used. (Lines 143-146)

Revised manuscript

To present the characteristics of reported suicide incidents and suicide decedents separately, two distinct datasets were created, with duplication from multiple newspaper reports on the same incident removed. Specifically, data from the earliest published article were selected; furthermore, if multiple reports on the same incident were published on the same day, the report from the newspaper with the largest circulation was used. (Lines 141-145)

Response 3-2-3

We adopted a consistent principle throughout the manuscript to present the dataset description, analyses, and results in the order of suicide decedents, incidents, and articles. The following is the revised explanation of the processes for constructing the incident and article datasets, revised in accordance with this principle.

Incident types primarily consisted of Famous person suicide, Homicide-suicide, and Multiple suicide. Incidents that did not fall under any of these types were categorized as unclassified type. When incident types were treated as a single categorical variable, those that fell under more than one type were classified as combination type.

Regarding suicide news articles, minor adjustments were made to prepare the data for analysis. To accurately rate incidents involving more than one suicide decedent, 2-value (0, 1) variables related to sex and age—Female decedent included and Younger decedent included (defined as a decedent equal to or younger than the median age)—were created. When suicide decedents used different suicide methods within an incident, the less frequent one was used operationally for classification. (Lines 146-154)

We also revised the order of presentation to follow the sequence of suicide decedents, incidents, and articles, as shown in the example below.

“reported incidents and suicide decedents” → “reported suicide decedents and incidents” (Line 190)

Response 3-2-4

The significance threshold (p < 0.025) was added to the descriptions of the results for the comparisons between admission numbers during the one-week periods before and after article release for each patient group, as well as for the correlation analyses between the Δs and article characteristics, to clarify that these sections pertain to article-level analyses to which the Bonferroni correction was applied.

The significance of the increase in admission numbers across all patient groups far exceeded the designated level (p < 0.025). …(Lines 336-337)

Weak but significant correlations (p < 0.025) between Δs for patient groups and article characteristics were as follows … (Lines 343-344)

Comment 3-3

SA patients were divided into "Younger" and "Older" groups based on whether they were below or above the median age of suicide decedents. While this choice is theoretically justifiable, especially if guided by identification theory, which emphasizes alignment between media representations (decedents) and recipients (patients), the rationale is not clearly explained in the current manuscript. I recommend that the Authors add a sentence explicitly clarifying that this cutoff was chosen not to reflect the patient population distribution, but rather to test age-related susceptibility to suicide reporting effects. In other words, the Authors should clarify this as a theory-driven choice.

Response 3-3

The distinction between younger and older groups of SA patients and reported suicide decedents was made to examine the association between patient age groups and the basic characteristics of suicide news articles, with particular attention to the age groups of the suicide decedents. Ideally, the analysis would be stratified into more detailed age groups; however, due to limited statistical power, we opted for this broad categorization as a pragmatic approach.

Response 3-3-1

The limitation that the age groupings were too broad to allow clear conclusions has been revised in the Limitations section as follows.

Previous manuscript

Fourth, the fine categorization of variables reduces the number of cases per group, which may affect the study's results by lowering statistical power and increasing the likelihood of type 2 errors. (Lines 612-614)

Revised manuscript

Fifth, some analyses may have been affected by limited statistical power. For example, fine categorization reduced group sizes, increasing the risk of type II errors, and broad age groupings of reported suicide decedents and SA patients constrained the clarity of conclusions. (Lines 621-624)

Response 3-3-2

Because the results regarding age grouping cannot be interpreted definitively, we replaced "absence" with "lack" in the sections below.

The lack of evidence supporting the identification theory in this study may be due to its focus on routine (low impact) suicide reports …(Lines 589-590)

In this study, the lack of significant findings supporting the identification theory—such as evidence of correspondence between the sex and age of reported suicide decedents and those of SA patients,… (Lines 645-647)

Comment 3-4

The Authors state that follow-up news reports were excluded in order to focus on the short-term impact of initial suicide coverage within a one-week period. While this decision aligns with their analytic design, it does not fully address the concern about cumulative exposure. A growing body of literature suggest that repeated exposure to suicide-related content (over time or by multiple sources) can compound risk, particularly among vulnerable individuals. By excluding follow-up reports and analysing only the first article per incident, the Authors may inadvertently underrepresent the real-world exposure profile of individuals who consume multiple media reports on the same suicide event. While I appreciate the clarity of the short-term focus, the ecological validity of the findings is limited. The Authors should consider explicitly acknowledging this limitation or potential future direction in the manuscript, as it would strengthen the transparency of the study’s conclusions.

Response 3-4

Our decision to exclude follow-up suicide reports was primarily intended to simplify the evaluation of the articles analyzed, while capturing the most impactful coverage by focusing on initial reports. However, this selection strategy inevitably excludes other types of reporting, such as follow-up coverage, and should be acknowledged as a limitation.

Response 3-4-1

The limitation arising from restricting the target suicide reports to initial newspaper articles has been added to the Limitations section as follows, and the numbering of the subsequent limitations has been adjusted accordingly:

Second, selecting only initial news articles for analysis to achieve a simplified data structure introduces some problems, including the inability to assess the cumulative effects of follow-up reports. Future research should aim to incorporate and analyze other types of coverage. (Lines 611-614)

Additional correction 1

We have revised the short title as follows.

“Suicide Reporting; Effects on Sex and Age of ER Patients” → “Effects of Suicide News Reporting on Sex and Age of ER Patients.” (Line 5 in Abstract)

Additional correction 2

In the revised Abstract, the discussion was focused on the findings from the MANOVA. The following is the opening sentence of the Discussion subsection."

This study demonstrates a differential impact of suicide news articles reporting gas poisoning, infrequently reported suicide methods, and firearm discharge, on SA patient admissions across sex and age groups. (Lines 47-49)

Additional correction 3

In the previous manuscript, the study period for the suicide report collection was set as “(the period) between April 1, 2012, and January 31, 2019.” However, since the number of SA patient admissions during the one-week periods before and after article publication needed to be assessed, the correct study period should have been “(the period) between April 7, 2012, and January 25, 2019.” This point has now been corrected:

“(the period) between April 1, 2012, and January 31, 2019” (Line 117) → “between April 7, 2012, and January 25, 2019” (Lines 116-117)

Additional correction 4

The following sections have been revised using semicolons to ensure consistency with other parts of the manuscript.

Significant differences by sex and age group were found, as follows: younger decedents (≤ 35 years) included a higher percentage of females; females were less represented among occupational or social status groups of Famous persons, Public workers, Police officers or SDF members, and Criminal or suspected persons, as well as in cases involving Hanging or Firearm discharge; in contrast, females were more represented in cases involving Height jumping and Gas poisoning.

Younger decedents accounted for a smaller percentage of Famous persons, Public workers, and Professional experts; in contrast, they accounted for a higher percentage in cases involving Railroad jumping and Height jumping, and a lower percentage in cases involving Hanging, Firearm discharge, and Cutting or stabbing. (Lines 261-269)

Significant associations between suicide incident types and sex- and age-related characteristics were observed, as follows: Female decedent included was negatively associated with Famous person suicide and positively associated with Multiple suicide; Younger decedent included was negatively associated with both Famous person suicide and Homicide-suicide. (Lines 282-285)

Additional correction 5

The first paragraph of the subsection "Associations between Reported Suicide Methods and Sex and Age Group Admissions" in the previous manuscript introduced studies that addressed the overall impact of reported suicide methods on news recipients, rather than the differential impact across sex and age groups. Therefore, it has been moved to the earlier subsection "Associations Between Newspaper Suicide Reports and SA Patient Admissions," and a discussion has been added regarding the findings presented there.

It was recognized that reporting on other suicide methods was not associated with changes in the overall number of admissions. By contrast, in previous reviews on suicide cases, (1, 5, 7, 11, 13, 15, 27) have reported that media coverage of specific suicide methods influenced the occurrence of suicides. Some studies have found that certain methods, such as jumping from a height, (8, 14, 23) are associated with an increase in suicide incidents, whereas others, such as cutting or stabbing, and railroad jumping,(23) may be associated with a decrease in suicide rates.

This discrepancy may be explained by differences in the research focus—specifically, whether the outcome examined was suicide or suicide attempt—and by variations in the socio-cultural backgrounds of the study settings

---

## [Decision Letter · Decision Letter 2]

31 Jul 2025

Sex and Age Differences in the Association between Routine Suicide Newspaper Reporting and Change in Admissions of Suicidal Patients: An Investigation at an Emergency and Critical Care Center in Tokyo

PONE-D-24-46119R2

Dear Dr. Hayashi,

We’re pleased to inform you that your manuscript has been judged scientifically suitable for publication and will be formally accepted for publication once it meets all outstanding technical requirements.

Kind regards,

Paolo Roma

Academic Editor

PLOS ONE

Reviewers' comments:

Reviewer's Responses to Questions

**Comments to the Author**

1. If the authors have adequately addressed your comments raised in a previous round of review and you feel that this manuscript is now acceptable for publication, you may indicate that here to bypass the “Comments to the Author” section, enter your conflict of interest statement in the “Confidential to Editor” section, and submit your "Accept" recommendation.

Reviewer #3: All comments have been addressed

Reviewer #4: All comments have been addressed

2. Is the manuscript technically sound, and do the data support the conclusions?

Reviewer #3: Yes

Reviewer #4: Yes

3. Has the statistical analysis been performed appropriately and rigorously? 

Reviewer #3: Yes

Reviewer #4: Yes

4. Have the authors made all data underlying the findings in their manuscript fully available?

Reviewer #3: Yes

Reviewer #4: Yes

5. Is the manuscript presented in an intelligible fashion and written in standard English?

Reviewer #3: Yes

Reviewer #4: Yes

6. Review Comments to the Author

Reviewer #3: (No Response)

Reviewer #4: Sex and Age Differences in the Association between Routine Suicide Newspaper Reporting and Change in Admissions of Suicidal Patients: An Investigation at an Emergency and Critical Care Center in Tokyo

Thanks for addressing the comments properly.

7. PLOS authors have the option to publish the peer review history of their article (what does this mean? ). If published, this will include your full peer review and any attached files.

**Do you want your identity to be public for this peer review?** For information about this choice, including consent withdrawal, please see our Privacy Policy .

Reviewer #3: No

Reviewer #4: No

---

## [Editor Report · Acceptance letter]

PONE-D-24-46119R2

PLOS ONE

Dear Dr. Hayashi,

I'm pleased to inform you that your manuscript has been deemed suitable for publication in PLOS ONE. Congratulations! Your manuscript is now being handed over to our production team.

Kind regards,

on behalf of

Prof. Paolo Roma

Academic Editor

PLOS ONE